# Contrasting Styles of Inter-Caldera Volcanism in a Peralkaline System: Case Studies from Pantelleria (Sicily Channel, Italy)

**Pierangelo Romano** [1], **John Charles White** [2], **Silvio Giuseppe Rotolo** [1,3,*], **Nina J. Jordan** [4], **Rosolino Cirrincione** [5], **Giovanni De Giorgio** [5], **Patrizia Fiannacca** [5] and **Epifanio Vaccaro** [6]

1. Istituto Nazionale di Geofisica e Vulcanologia (INGV), Sez. di Palermo, 90146 Palermo, Italy; pierangeloromano@gmail.com
2. Department of Physics, Geosciences, and Astronomy, Eastern Kentucky University, Richmond, KY 40475, USA; john.white@eku.edu
3. Dipartimento di Scienze della Terra e del Mare (DiSTeM), Università degli Studi di Palermo, 90123 Palermo, Italy
4. School of Geography, Geology and the Environment, University of Leicester, University Road, Leicester LE1 7RH, UK; nina.jordan@rwth-aachen.de
5. Dipartimento di Scienze Biologiche, Geologiche e Ambientali, Università degli Studi di Catania, Corso Italia 57, 95129 Catania, Italy; rosolino.cirrincione@unict.it (R.C.); gidegiorgio@icloud.com (G.D.G.); patrizia.fiannacca@unict.it (P.F.)
6. The Natural History Museum, Cromwell Road, London SW7 5BD, UK; e.vaccaro@nhm.ac.uk
* Correspondence: silvio.rotolo@unipa.it; Tel.: +39-091-23861608

**Abstract:** The recent (<190 ka) volcanic history of Pantelleria is characterized by the eruption of nine peralkaline ignimbrites, ranging in composition from comenditic trachyte to comendite to pantellerite. The ~46 ka Green Tuff (GT) was the last of these ignimbrites, which was followed by many effusive and explosive low-volume eruptions of pantellerite from vents within the caldera moat and along the caldera rim. Although recent studies have shed additional light on the age, petrochemistry, and volcanology of the older ignimbrites, there is very little knowledge of magmatism that occurred between these older ignimbrites, primarily due to the very scarce exposures. In this paper, we present new field descriptions and geochemical data for three local peralkaline centers never studied before, two pre-GT and one post-GT, which share a similar setting with respect to the caldera scarps but differ in terms of their age, composition, and eruptive style. These centers include: (i) the older (~125 ka) Giache center (comenditic trachyte), (ii) the ~67 ka Attalora center (comendite, pantellerite), and (iii) the younger (~14 ka) Patite center (pantellerite).

**Keywords:** caldera; peralkaline; pantellerite; Pantelleria

## 1. Introduction

The geochronologic history of the island of Pantelleria can be divided into three major phases [1], without any implication for either petrogenetic or volcanic relationships [2], given that within each of these phases there are major stratigraphic breaks:

(i) The first phase (~324–190 ka) consists of effusive and explosive activity extensively buried by younger deposits and exposed exclusively along the remote south coast; there are few geochronological or geochemical data available for rocks from this phase.

(ii) The second phase (~190–46 ka) includes the eruption of nine ignimbrites, ranging in composition from comenditic trachyte to pantellerite [2,3]. The collapse of the "La Vecchia" caldera occurred during this second phase, and it is dated at ~140–146 ka [1]. The younger, partially nested inner "Cinque Denti" caldera has frequently been interpreted to have formed by the eruption of the final Green Tuff ignimbrite [4] at 46.7 ka [5]; however, at least some parts of this structure probably predate the Green Tuff [6,7]. In the time between each ignimbrite eruption (8–45 ka age intervals), several effusive and explosive eruptions took place from small local centers that built pumice cones and/or lava domes/shields,

in a scenario likely similar to the present-day post-Green Tuff distribution of more than 40 scattered effusive and explosive (or both) volcanic centers [4].

(iii) The third and final phase began after the Green Tuff eruption, with the emplacement of a voluminous trachyte lava body from the monogenetic intra-caldera Monte Gibele vent (including the Montagna Grande uplift), which was followed by a spatially and temporally diffuse period of effusive and mildly explosive activity to ~6–8 ka [8,9] that resulted in a complex series of interdigitated pantelleritic pumice fall sequences [10,11]. This mildly explosive activity was centered mostly on or within the Cinque Denti caldera rim and alternated with pantellerite lava flows that erupted contemporaneously with basaltic lavas (scoria cones, lava flows), which are limited to the northern portion of the island. In a comprehensive approach, the whole volcanological history of Pantelleria can be considered as an alternation of (nine) ignimbrite events and inter-ignimbrite periods punctuated by the activation of several minor and scattered volcanic centers [2].

While there have been many studies (petrographic, volcanological, geochronological, paleomagnetic) on the ignimbrites and the widespread volcanism that followed the Green Tuff, there is scant knowledge of inter-ignimbrite volcanism. To fill this gap in knowledge, we have chosen three representative eruptive centers that have been very poorly studied: two in the south sector of the island (Cuddia Attalora and Cuddia Patite) and one in the east sector (Cala delle Giache). These centers differ in age, composition, erupted volumes, and eruptive styles, but they share a position in proximity to the (older) La Vecchia caldera buried annular fault system (Attalora and Giache centers) or to the (younger and inner) Cinque Denti caldera one (Patite center), suggesting that pre-existing caldera faults may have served as possible pathways for pantellerite magma ascent.

The principal aims of this paper are thus: (i) to describe the petrography and field relations of these poorly known eruptive centers and their erupted rocks, and (ii) to determine pre-eruptive magma storage conditions and magma evolution, with possible inferences on magma productivity in inter-ignimbrite periods, over a temporal range of 53 ka, i.e., the age interval separating activity of the Patite (age 14 ka) and Attalora (age 67 ka) centers, which likely shared a common plumbing system, and ≥50 ka, the age difference between the two older centers Attalora and Giache, both located very close to the La Vecchia caldera annular faults.

## 2. Geologic Setting

The island of Pantelleria is situated in the Sicily Channel Rift Zone (Figure 1), a thinned continental domain belonging to the Pelagian Block, affected since the Pliocene by transtensional tectonics and diffuse subaerial (Linosa and Pantelleria islands) and submarine volcanism. At Pantelleria, the type locality of pantelleritic rocks (peralkaline rhyolites enriched in Na, Fe, Cl), the majority of exposed rocks are felsic, ranging in composition from trachyte to pantellerite, while mafic magmatism has only occurred in the northern sector of the island, where it has formed scoria cones and lava flows, as well as the 1891 CE submarine cone 5 km off the island [7]. The new eruptive history proposed by Jordan et al. [2] defined a total of nine ignimbrites erupted between ~190 to 46 ka, which alternated between a multitude of local eruptive centers (Figure 2). The ignimbrites are typically welded and variably rheomorphic, and they are frequently associated with pumice fallout deposits, located either below or atop the main ignimbrite body [2]. Five of the nine ignimbrites also include local polylithic breccias, variably welded, that might be interpreted as evidence of caldera collapse episodes, possibly along reactivated structures [2], although they may also simply represent a major phase of vent widening during a climactic phase of the eruption. The total onshore volume (not taking in account the amount of tephra dispersed in the sea) of all nine ignimbrites, including the Green Tuff, sums to ~2.3 km$^3$ DRE, with individual volumes ranging from 0.003 to 0.64 km$^3$ DRE, with an apparent decrease in eruption size from 187 to 85 ka and with a slight increase for the 45.7 ka Green Tuff (GT). Another major conclusion from Jordan et al. [2] is the apparent remarkable similarity in volcanological evolution among all the inter-ignimbrite periods in a scenario comparable

to the present-day distribution and type of local eruptive centers formed after the Green Tuff ignimbrite eruption. Post-GT volcanism has been by far dominated by pantelleritic magmatism, which erupted either (i) purely effusively (e.g., the pantellerite lava centers of Gelkhamar, Gibele-M.gna Grande, etc.), (ii) purely explosively (strombolian eruptions, e.g., Gadir, Tikirriki, Patite), or (iii) in both ways (e.g., Attalora, Giache, Randazzo, etc.) [7].

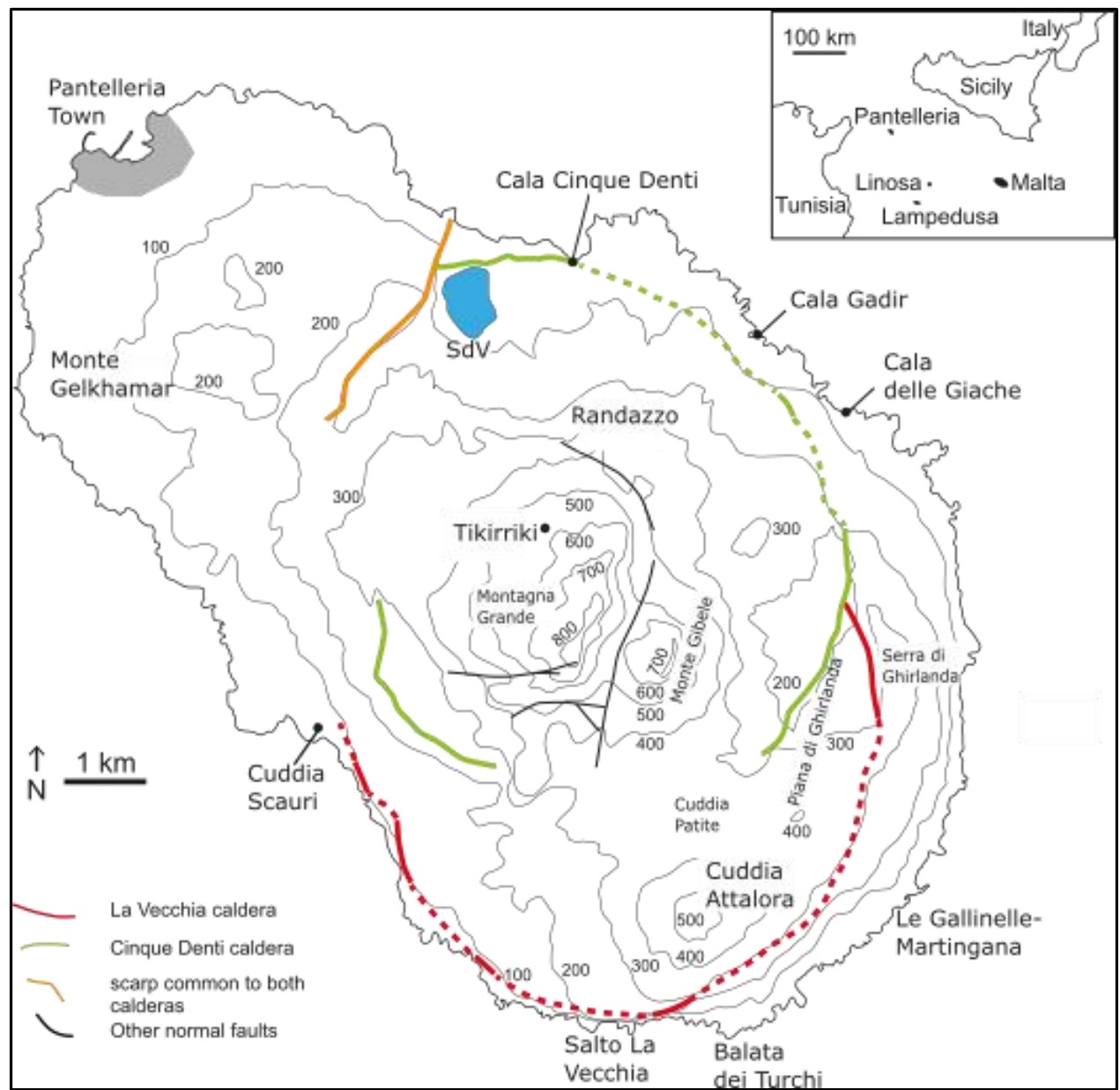

**Figure 1.** The island of Pantelleria with locations mentioned in the text (SdV, lake Specchio di Venere). Inset: location of Pantelleria within the then Sicily Strait. Green and red dashed lines refer to inferred caldera rims.

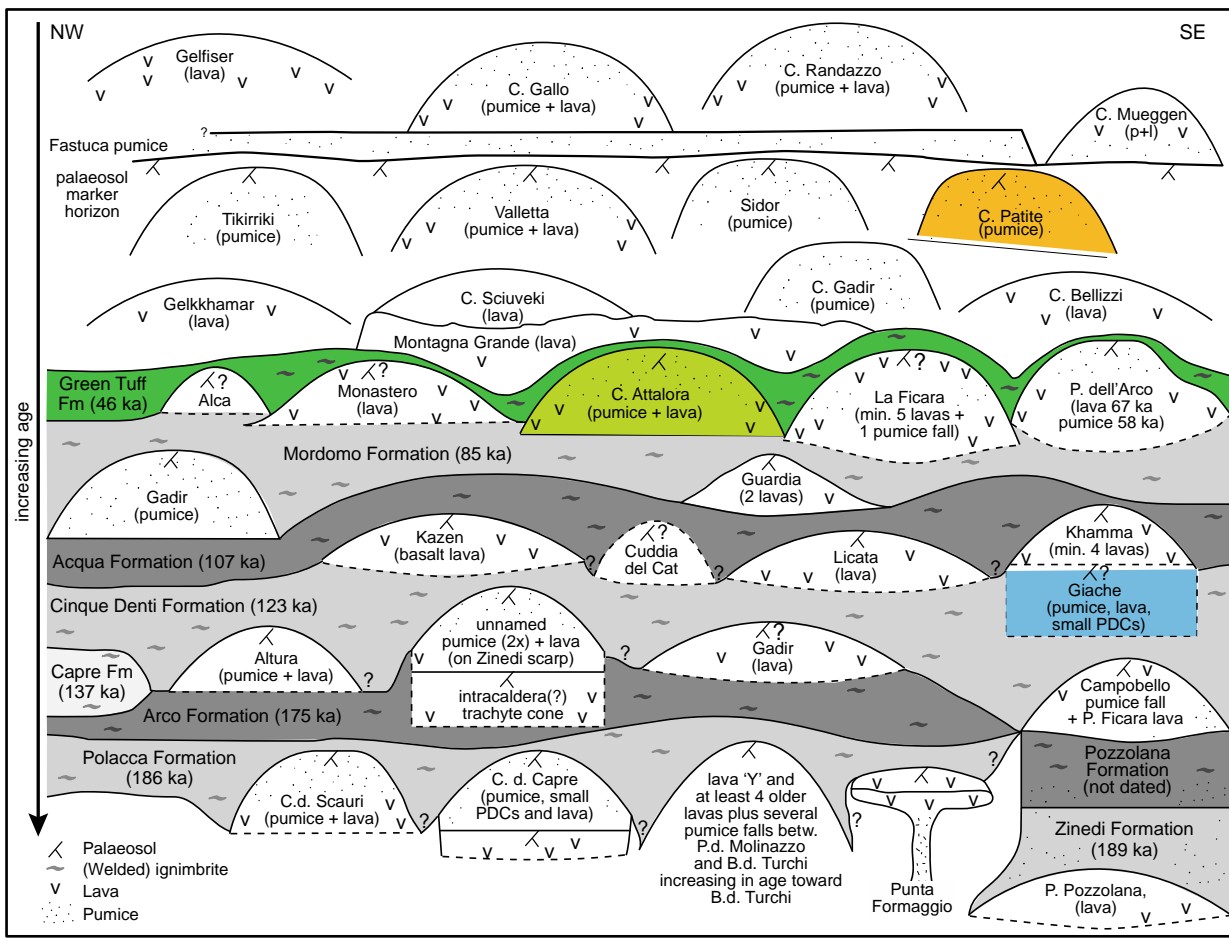

**Figure 2.** Schematic chronological representation of Pantelleria volcanological evolution (ignimbrite formations and local centers) pre- and post-Green Tuff (modified from [1]). Eruptive centers studied in this work are shown colored: Giache (blue), Cuddia Attalora (yellow), and Cuddia Patite (orange). Adapted from [2].

## 3. Stratigraphy of the Volcanic Centers

Because of the extensive cover of ignimbrite sheets, inter-ignimbrite centers are visible only in the high, steep coastal scarps that characterize the southeastern to western coast of Pantelleria. In particular, the southern sector of the island (Figure 1) offers the opportunity to study two centers positioned 1 km from each other (Patite, Attalora), possibly sharing a similar magma source (and plumbing system), despite being separated by an age difference of ~53 ka:

- The pre-Green Tuff *Cuddia Attalora* (*Cùddia* is a local name that translates to "hill"), an explosive and effusive volcano, which represents the largest of the pre-Green Tuff inter-ignimbrite centers, is located 600 m north of the buried rim of the old caldera (La Vecchia caldera, age 140–145 ka).
- The post-Green Tuff *Cuddia Patite* center, a dominantly explosive volcano, is situated immediately above the buried rim of the younger caldera (Cinque Denti caldera, age ≥ 46 ka).
- The *Cala delle Giache* center, a pre-GT monogenetic volcano located in the eastern sector, is situated 300 m east of the buried rim of the older (La Vecchia) caldera, which is exposed as a perfect longitudinal cross-section on the east coast, having a peculiar assemblage of a pumice fallout, a very small-volume ignimbrite, as well as a 150 m × 30 m sub-volcanic intrusion with its well-exposed feeding dike.

### 3.1. Cuddia Attalora Center (ATT)

Cuddia Attalora (elevation 560 m a.s.l.) represents the largest volcanic center after the M. Gibele-M.gna Grande post-GT lava complex and is, thus, by far the largest of the inter-ignimbrite centers. It stratigraphically overlies the Mordomo ignimbrite ($^{40}Ar/^{39}Ar$ age = 85 ka) [1] and is covered by the Green Tuff ($^{40}Ar/^{39}Ar$ age = 45.7 ka) [5], consistent with its single K/Ar age of $69 \pm 9$ ka [4] (Figures 2 and 3).

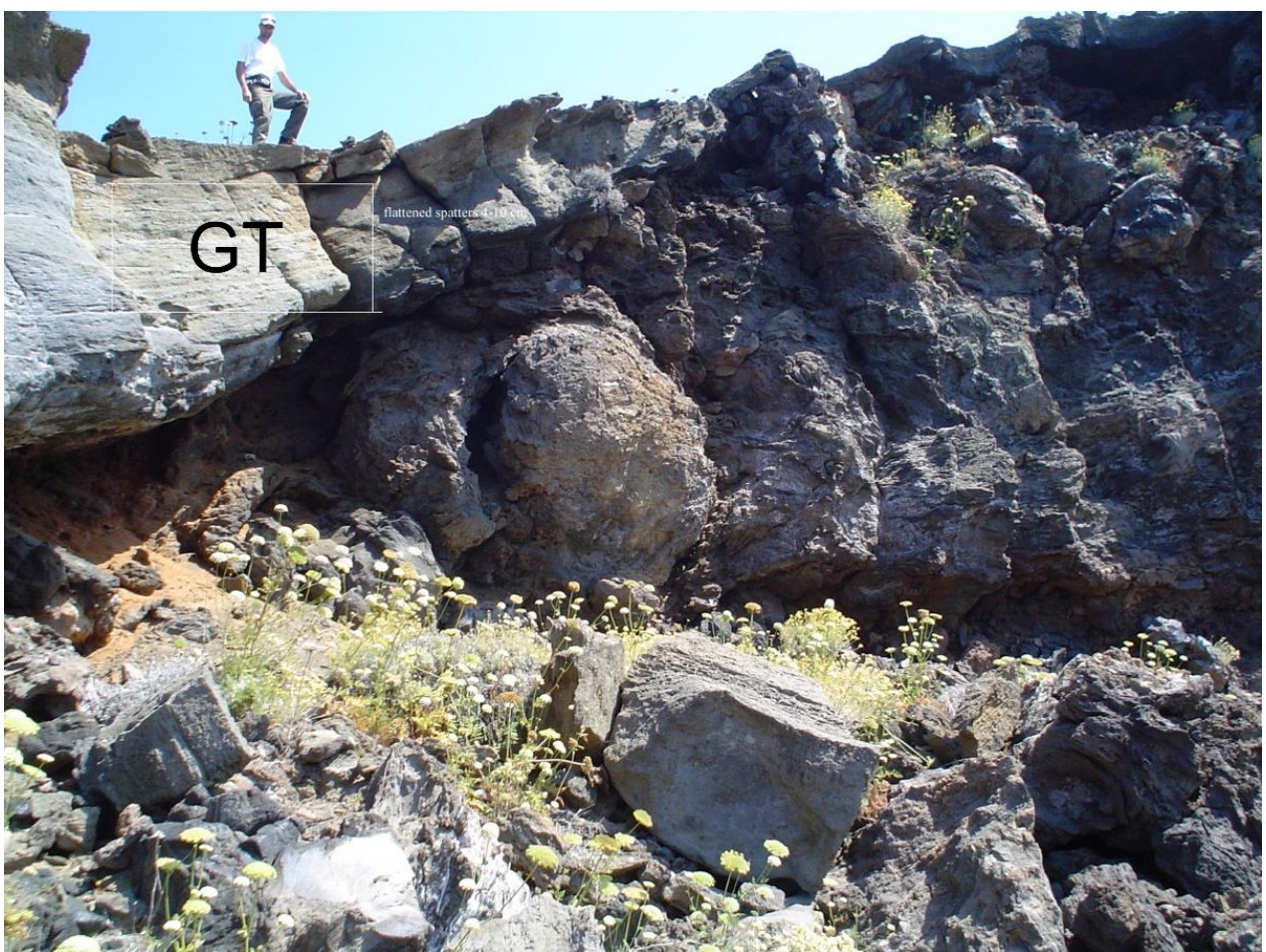

**Figure 3.** Attalora lava below the Green Tuff (GT) ignimbrite just east of Balata dei Turchi.

The activity of the Cuddia Attalora volcano began with an explosive eruption that produced a pantellerite pumice fall deposit, which was followed by the emission of radially directed pantellerite lava flows. The complete succession is visible near the top of the Salto La Vecchia cliff and at Balata dei Turchi (Figure 1). The pumice fallout succession has a maximum thickness of 22 m, as visible in the west side of Balata dei Turchi (Figure 1), and it consists of three units, from bottom to top (Figures 3 and 4):

(i) A 2 m thick well-sorted basal pumice layer, with highly vesicular pumice clasts and a mean pumice diameter of 2 cm (3 cm maximum). Crystal content is $\leq$15 vol% and consists dominantly of alkali feldspar (length $\leq$ 2 mm) with subordinate clinopyroxene > olivine > opaques. Dark hyalopantellerite lithics are frequent, with a diameter in the range of 1–2 cm.

(ii) An 8–10 m thick, highly vesicular middle unit with coarse (up to 8 cm) pumice. Crystal content is $\leq$15 vol% (alkali feldspar > clinopyroxene > olivine > opaques). The top of both the lower and middle units are characterized by an enrichment in pantellerite lithic fragments.

(iii) A 12 m thick upper unit (Figure 5), which is less vesiculated than the underlying units. It consists of rather well-sorted pumices up to 7 cm in diameter and a crystal content of ≤15 vol% (alkali feldspar > clinopyroxene > olivine > opaques).

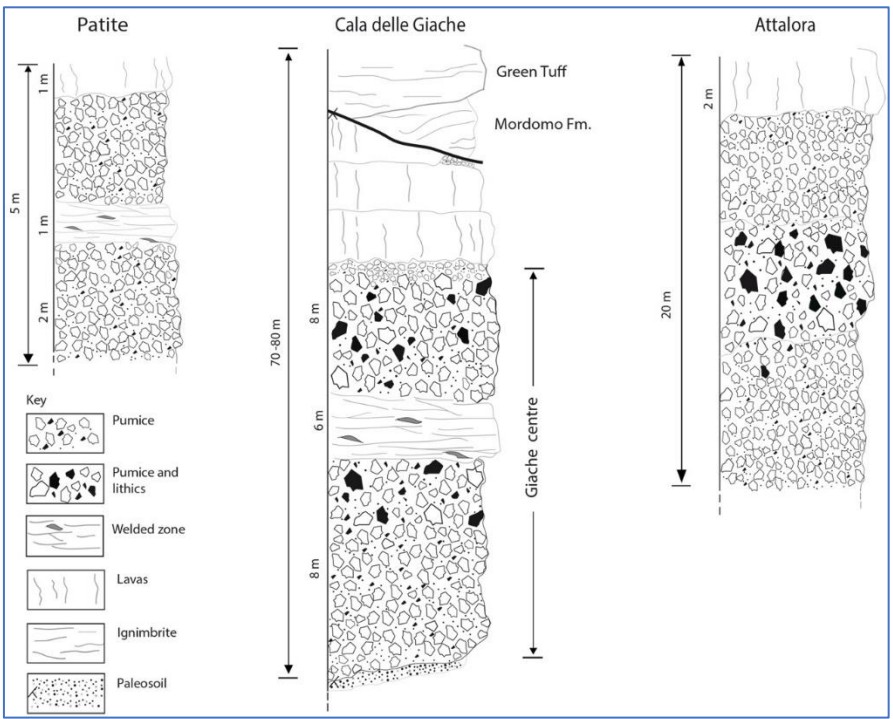

**Figure 4.** Simplified logs of the three local peralkaline centers studied in this work. Further details in the text.

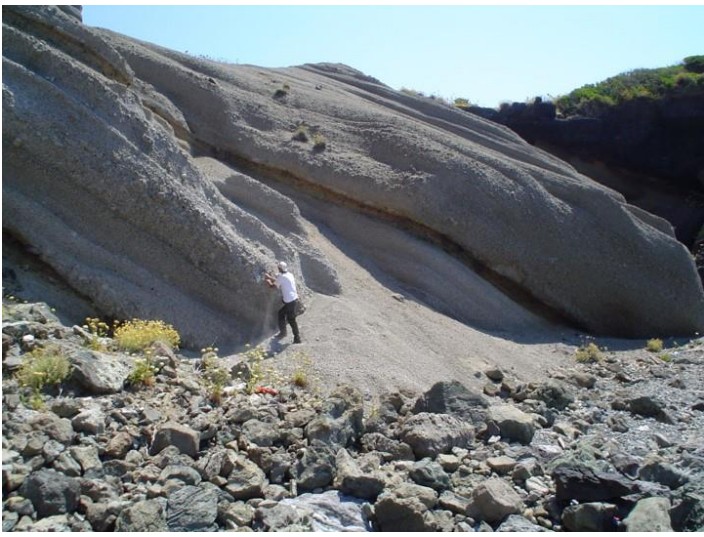

**Figure 5.** Attalora upper pumice, slightly west of Balata dei Turchi.

The Attalora fall succession is capped by a vitric pantellerite lava (Figure 2) that crops out at three distal sites below the Green Tuff:

(i) Balata dei Turchi (east side), 1.5 km south from the inferred vent, at a height of 5 m. a.s.l., directly below the GT with a thickness around 4 m.

(ii) Le Gallinelle, in Martingana, 2.5 km east from the vent.

(iii) The base of Piano di Ghirlanda scarp, 3 km NNE from the vent (Figure 1), just above the Attalora fall pumice.

### 3.2. *Cala Delle Giache Center (GIA)*

The Cala delle Giache center is located 300 m east from the La Vecchia caldera rim on the NE coast of Pantelleria. It lies below the ~107 ka Acqua ignimbrite and is thought to have produced the lavas forming Punta Zinedi at one end of the exposure. These have been dated to 124 ± 6 ka (K/Ar [4]).

The 200 m × 90 m coastal exposure at Cala delle Giache (40 m of which occurs at the Giache center, Figure 6) shows a whole range of eruption styles as inferred from the diversity of the deposits, including pumice fall, small scale pyroclastic density currents (PDCs), a second episode of pumice fall, and final lava effusion (Figure 4). A lower pumice package consists of alternating layers of large (10–30 cm) clast-supported pumice with subordinate lithics (Figure 7) and laterally discontinuous, locally cross-bedded fine layers with clasts ranging from 2 mm to ~2 cm. We interpreted this to result from a dominant fallout with intermittent PDCs. Higher in the succession, there is a welded cross-bedded zone consisting of dominant ash and minor lapilli juvenile fragments of pantellerite composition, up to ~1 cm (Figure 8), interpreted as a PDC deposit. Since it is fully welded, it may record an increase in mass flux or eruption temperature. On top of this, there is a thick welded layer with parallel bedding, i.e., a welded fall deposit. It transitions upward into nonwelded clast-supported pumice relating to a waning mass flux or temperature. The above-described deposits are capped by a number of lavas (inaccessible high up in the cliff); one is visible at the right of Figure 6.

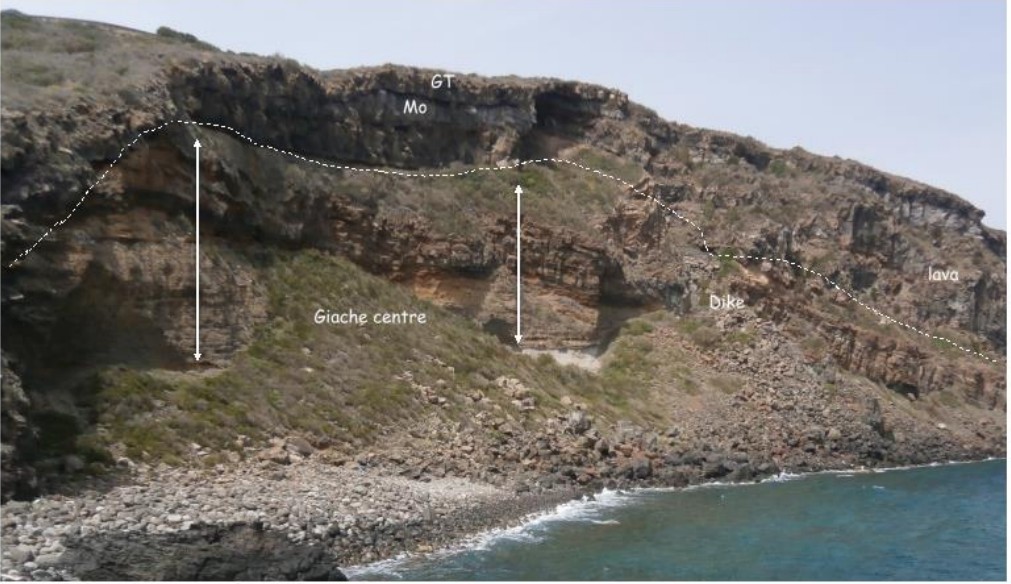

**Figure 6.** Giache center seen from a distance (view north-west); GT = Green Tuff ignimbrite, Mo = Mordomo Fm. Ignimbrite.

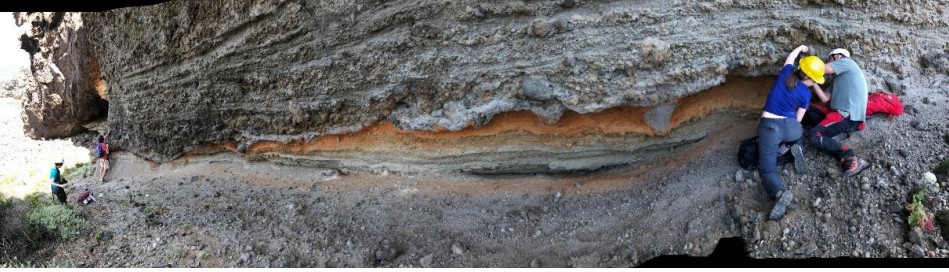

**Figure 7.** At Cala delle Giache, the base of the deposits related to the Giache local center is visible just above the paleosol and begins with a lithic-rich pumice fall deposit. Below the paleosol, there is instead visible a fine-grained uncorrelated phreatomagmatic fall deposit.

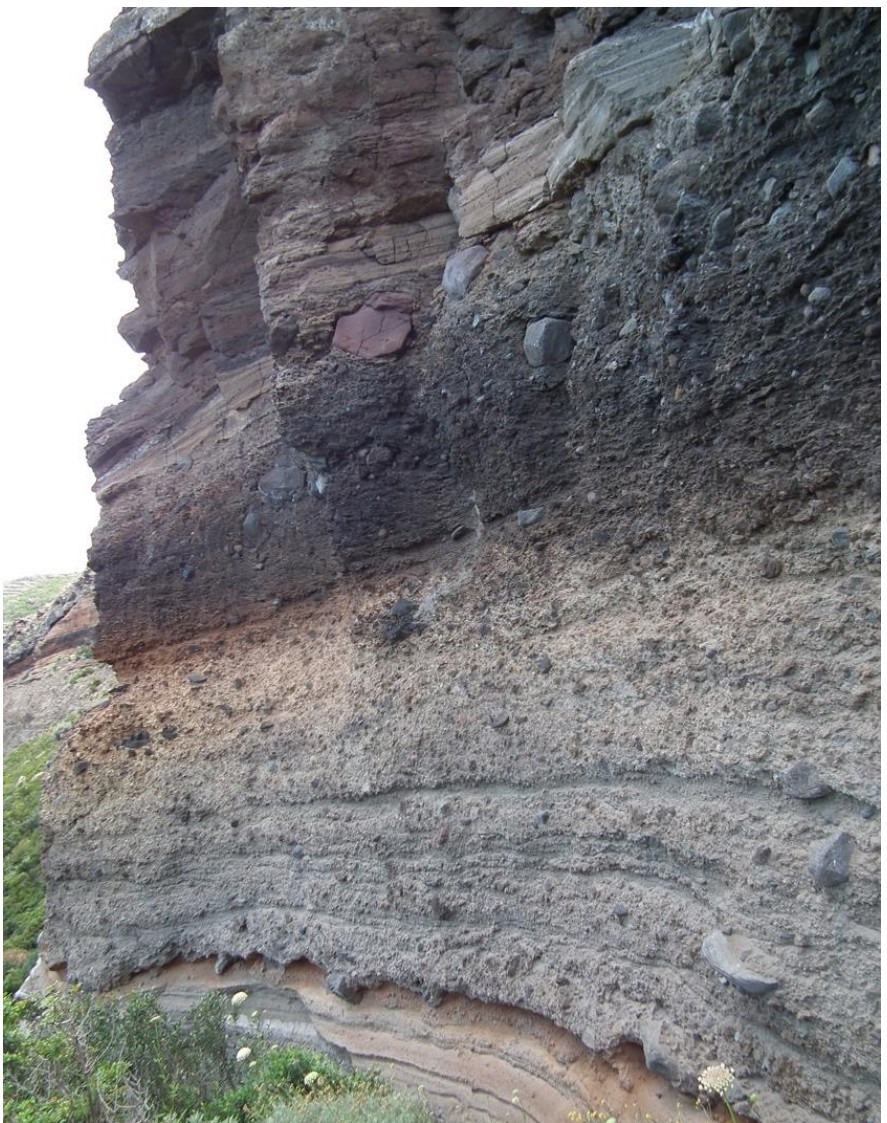

**Figure 8.** Cala delle Giache: detail on the transition from the base of the deposit with pumice resulting from dominant fallout and subordinate PDC deposit (gray). The darker area higher up indicates a transition to welding. The brown areas at the top are fine-grained, cross-bedded, densely welded and relate to pyroclastic density currents.

Acqua ignimbrite consists of a small outcrop, barely visible at the top left below Mordomo Fm.

### 3.3. Cuddia Patite Center (PAT)

The activity of the 14.4 ka Cuddia Patite center [8], 1 km north of the Attalora volcanic center (Figure 9), was initially explosive with the eruption of a thick pantellerite pumice fall, followed by the emission of short pantellerite lava flows. The pumice fall is widely dispersed in the area between Cuddia Attalora and Serra Ghirlanda (Figure 1). Near-vent pumices are coarse grained (Figure 10) and poorly sorted with a maximum clast size of up to 20 cm in diameter, with rare near-vent vitrophyric agglutinates. The phenocryst content is ~10 vol% with alkali feldspar (up to 2 mm in length), which is distinctly more abundant than quartz > aenigmatite > clinopyroxene.

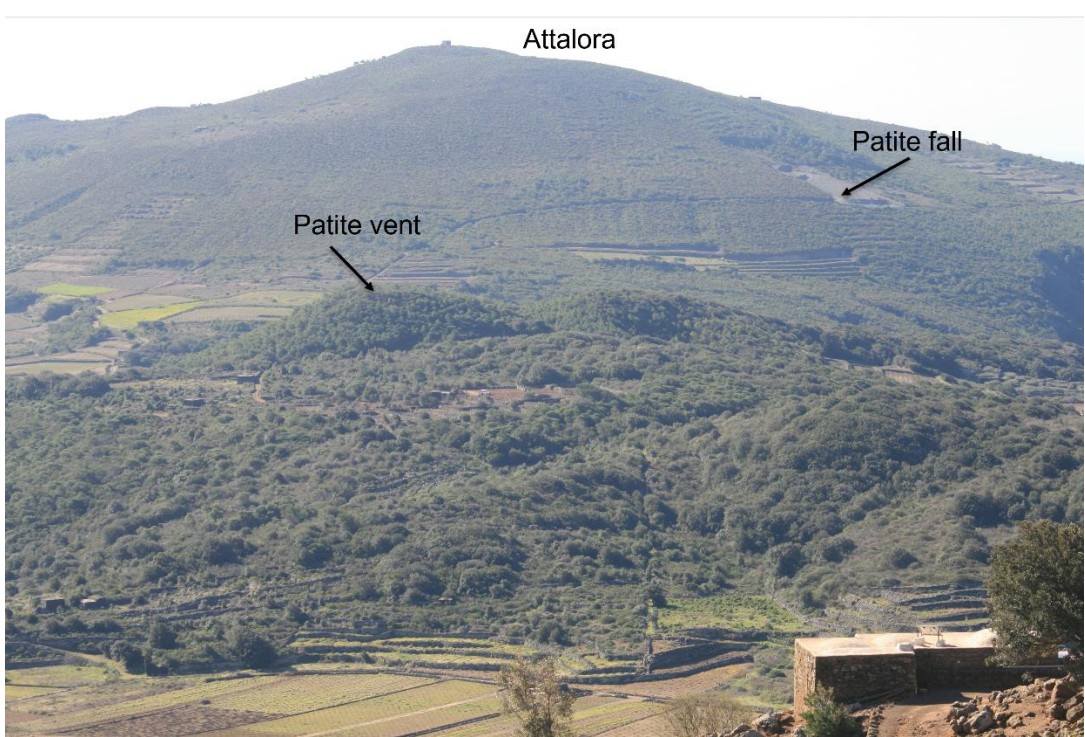

**Figure 9.** The post-Green Tuff Patite center adjacent (~1 km north) to the pre-GT Cuddia Attalora (view south).

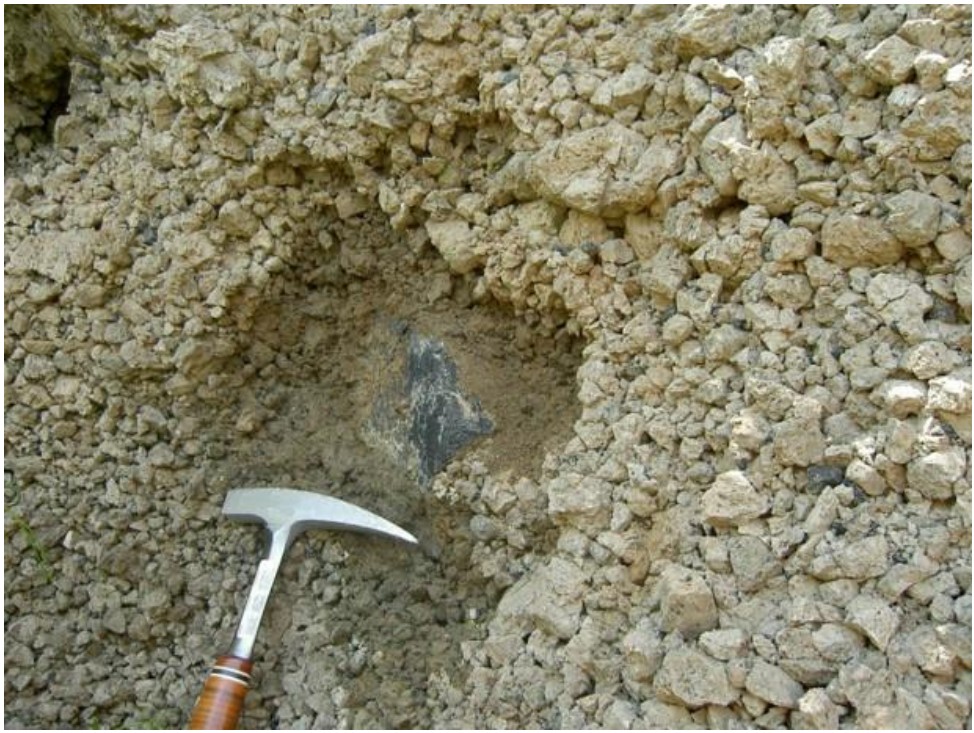

**Figure 10.** Patite pumice fall deposits in a section 150 m west of the vent.

## 4. Analytical Techniques

We processed 18 selected samples (four lavas and 14 pumices) for major and trace element whole rock analysis (Supplementary Table S1) and mineral chemistry (Supplementary Tables S2–S5). Pumice and lavas from Attalora and Patite were first left overnight in distilled water to minimize the effects of marine aerosol adsorption. Whole-

rock powders were analyzed for major and trace elements by wavelength-dispersive x-ray fluorescence spectroscopy (WD-XRF) with a RIGAKU ZSX-Primus housed in the Dipartimento di Scienze della Terra e Mare (DiSTeM) at the University of Palermo. Samples from Giache were analyzed on a PW4400 Axios WD-XRF with a 4-kW rhodium tube at the University of Leicester (see [3,12] for details).A chip of each rock sample was reduced to a polished thin section or mounted in epoxy for SEM-EDS microanalyses. To reach a sufficiently representative number of phenocrysts in pumice samples, mineral concentrates (size class 0.125–0.250 mm) were also mounted in epoxy for SEM-EDS work. Mineral and textural analyses were performed using an OXFORD LEO™ 440 Scanning Electron Microscope coupled to an Oxford-Link EDS, at DiSTeM. Operating conditions were: 20 kV accelerating voltage and 600 pA beam current. Natural mineral standards were used to calibrate quantitative analyses. Giache samples were analyzed at the University of Leicester on a JEOL JXA-8600S with an accelerating voltage of 15 keV and 20 s counting time (10 s for background) (see [12] for details).

## 5. Results

### 5.1. Classification and Major Element Geochemistry

All rock samples from the Attalora, Giache, and Patite volcanic centers are peralkaline (peralkalinity index, P.I. > 1.0; Figure 11a) and are further classified following IUGS guidelines [13,14]. Samples from the Attalora rocks are comendite and pantellerite; those from Giache are comenditic trachytes that plot near the boundary with pantelleritic trachyte; Patite samples are pantellerite (Figure 11c).

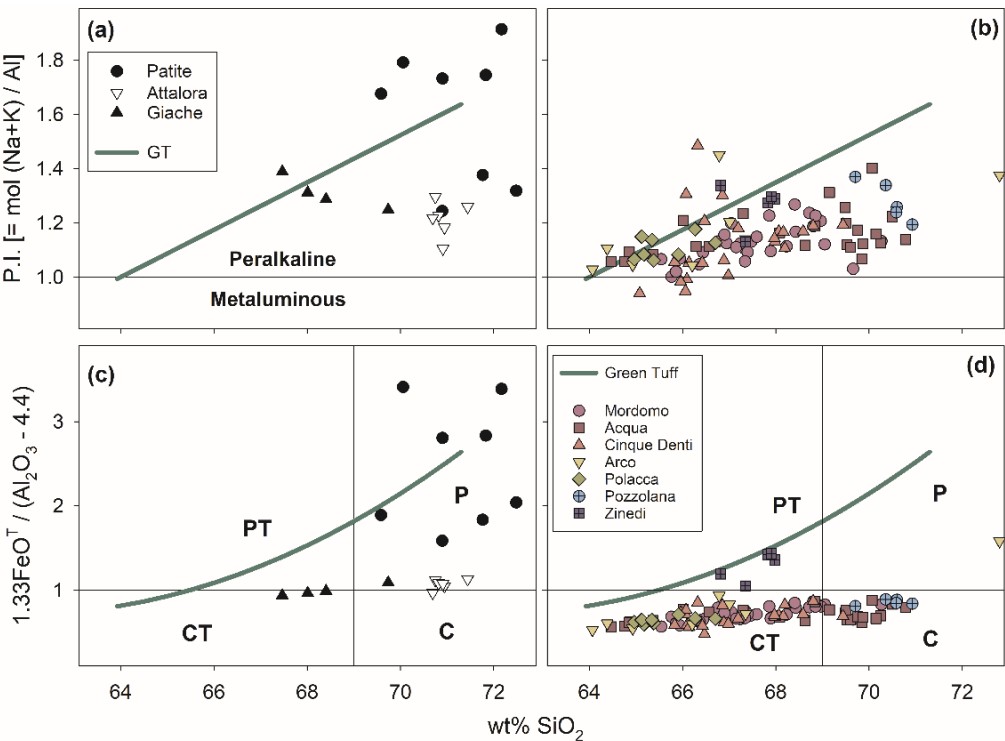

**Figure 11.** Classification of the volcanic rocks from the Giache, Attalora, and Patite (**a,c**) volcanic centers in comparison with the Green Tuff trend (**b**) [15] and pre-Green Tuff ignimbrites (**d**) [3]. P, Pantellerite; C, Comendite; PT, Pantelleritic Trachyte; CT, Comenditic Trachyte.

For comparison with the caldera-forming ignimbrites, Figure 11 also includes the Green Tuff trend [15] and the data from the pre-Green Tuff ignimbrites [3].

Because most of these analyzed samples consist of pumice, the loss on ignition (LOI) value is consistently high and there is considerable scatter in the trace element data. Therefore, in this study, we focused on the petrography and mineral chemistry from the oldest center to the youngest.

### 5.2. Petrography and Geochemistry

Pantellerite and borderline-comendite samples from the Attalora volcanic center consist of lavas and pumice dominated by unzoned, euhedral to subhedral phenocrysts of anorthoclase to sanidine ($Or_{31-38}$), with subordinate clinopyroxene, olivine, ilmenite, and magnetite. Clinopyroxene occurred as both inclusions in anorthoclase and as microlites and ranges from augite to sodian augite (1.2–2.4 wt% $Na_2O$, $En_{23-15}Wo_{41-43}$). Olivines occurred as inclusions in both anorthoclase and clinopyroxene and were more iron- and Mn-rich than those at Giache (83–91 mol% Fa, 2.7–6.0 wt% MnO). Ilmenite compositions were Ti-rich and typical of pantellerites (97–98 mol% Ilm). Three magnetite grains had highly variable compositions and had likely been significantly oxidized. Although temperature cannot be determined from oxide pairs, olivine-clinopyroxene equilibrium between the highest-Fe analyses suggested a temperature of ~742 °C at 1000 bars of pressure [16].

Comenditic trachyte samples from the Giache volcanic center consisted of pumice clasts of fallout origin set in a highly vesicular, devitrified matrix. The dominant phenocryst (~10 vol%) was euhedral to broken anorthoclase ($Or_{33}$), with minor clinopyroxene and fayalite and trace amounts of both ilmenite and magnetite. The clinopyroxene was low-sodium (<1.5 wt% $Na_2O$) augite with a composition ranging between $En_{26}Wo_{41}$ and $En_{15}Wo_{41}$ (recalculated following Lindsley and Frost [17]). Olivines were iron-rich (81–86 mol% Fa). Magnetite ($Usp_{65}$) and ilmenite ($Ilm_{96}$) (recalculated following Anderson [18]) were in equilibrium according to the criteria of Bacon and Hirschman [19] and yielded a temperature of 788 °C and $-16.4 \log fO_2$ (FMQ-1.6) [20,21]. This assemblage and temperature is typical of "type 1" comenditic trachytes at Pantelleria [22,23].

Pantellerite samples from Patite were amongst the most evolved samples from Pantelleria [22]. The assemblage contained no oxides or olivine and consisted only of dominant anorthoclase to sanidine ($Or_{33-38}$) and quartz, with minor sodian augite and aegirine-augite (2.6–4.0 wt% $Na_2O$, $En_{5-4}Wo_{46-53}$) and aenigmatite. Alkali feldspar phenocrysts range in size from 1–5 mm and are commonly euhedral to slightly rounded. Clinopyroxene is frequently enclosed as subhedral individuals in anorthoclase phenocrysts, but it also occurs as discrete phenocrysts. Aenigmatite occurs both in the groundmass and as inclusions in both alkali feldspar and clinopyroxene, indicating early crystallization.

Limited trace element data are available for samples from these three volcanic centers, but what does exist provides us with some possibly interesting insights into the origins of these magmas. Trace element variation diagrams that use Rb as a differentiation index are plotted in Figure 12. Despite being a typical "compatible" element, Ni (Figure 12a) increases with increasing Rb, which suggests no role for olivine in any of the fractionation sequences. High-field strength elements (HFSE) Y and Nb (Figure 12b,c) also show intrasuite increases with increasing Rb, as would be expected for incompatible elements, but samples from Giache have higher concentrations of each at low-Rb, strongly suggesting they belong to a liquid line of descent (LLOD) from a different basaltic parent enriched in HFSE [24]. Curious also is the increase in Ba with increasing Rb, which (coupled with the very low concentrations of Sr, not shown) may indicate that these magmas evolved via fractionation of plagioclase rather than alkali feldspar (see [25]). La and Ce (Figure 12e,f) are geochemically very similar, with a nearly constant ration (La/Ce = 0.46 $\pm$ 0.4) and behave incompatibly as expected relative to each other and Rb.

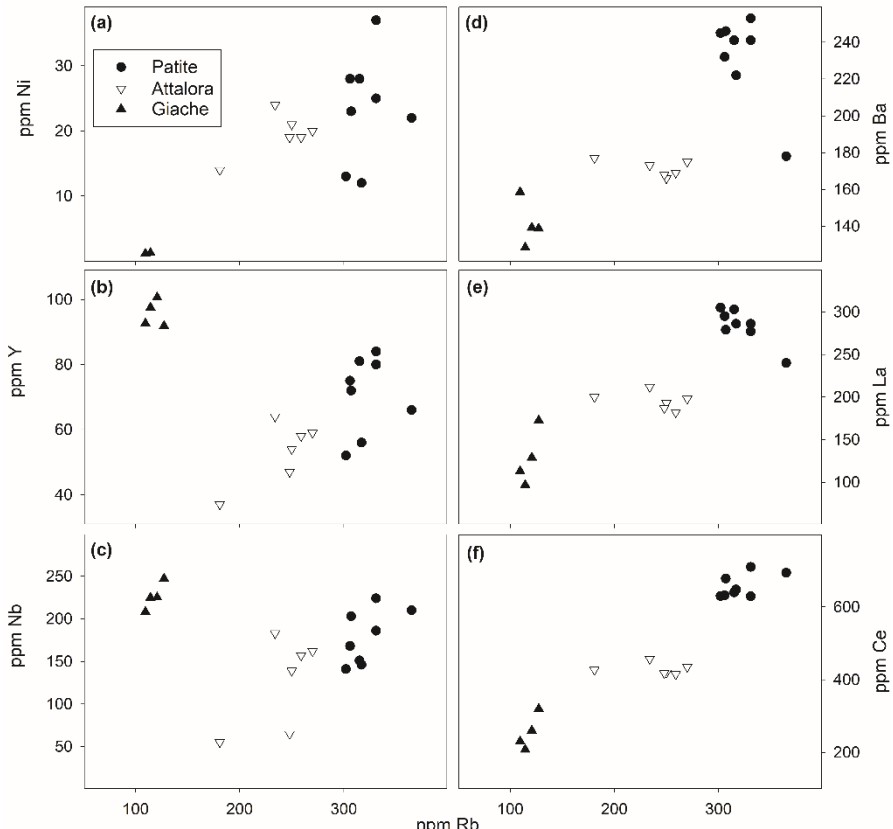

**Figure 12.** Trace element variation diagrams with Rb as the differentiation index. (**a**) Ni is a compatible element in olivine and has a positive correlation with Rb suggesting a lack of fractionation of this phase. (**b,c**) Y and Nb are incompatible elements in most minerals; the similarity between the Patite and Attalora trends suggest a similar petrogenetic origin, whereas the elevated values in Giache suggest a different petrogenetic origin. (**d**) The increase in Ba, compatible in alkali feldspar, may suggest fractionating of plagioclase rather than alkali feldspar. Similar trends with respect to the light rare earth elements (**e**) La and (**f**) Ce indicate the parental magmas to these centers were derived from a similar mantle source.

## 6. Discussion

A consistent number of papers on experimental petrology [23,26], petrological modeling [15,22,24,27], and fluid/melt inclusions [28–30] have been established at Pantelleria for pantellerite magma-fed eruptions, broadly similar to pre-eruptive magma conditions of 1.0–1.5 kbar pressure, temperatures of 700–750 °C, and water-rich (3.5–4.5 wt% $H_2O_{melt}$) conditions. A comparison of experimental data with geothermo-oxybarometric models and mineral associations [21,22] have recognized consistent sets of mineral assemblages that define a clear relationship between peralkalinity, water content, temperature, and oxygen fugacity. For instance, the assemblage *fayalite + ilmenite + magnetite + augite* characterizes pantellerites with relatively low peralkalinity (P.I. < 1.4), $fO_2$ around $\Delta$FMQ-1, and relatively high temperature (900 °C). The assemblage *aenigmatite + aegirine-augite + quartz* instead characterizes evolved pantellerites, with higher peralkalinity (P.I. > 1.8), higher $fO_2$ FMQ = 0), and lower temperature, ca. 700 °C [22,23]. Since textural evidence of crystal/melt disequilibrium were not noticed, it appears that our data extend the stability of the *fayalite + ilmenite + magnetite + sodian augite* assemblage up to a peralkalinity index of 1.6. The assemblage *aegirine − augite + quartz* is instead stable over a wide range of peralkalinity (P.I. = 1.5–1.9), below the value suggested by White et al. [22].

According to White et al. [22], the observed mineral assemblages of these pantelleritic samples are compatible with redox conditions between $\Delta$FMQ-1.5 and -0.5 for a temperature in the range 794–991 °C. Considering this $fO_2$ range, we can further constrain the magma

temperature from olivine composition by using the equation of Romano et al. [23], which relates fayalite content in olivine with $fO_2$, T, and $H_2O_{melt}$. As reported by several studies [27–31], melt water content in pantellerite ranges between 2.5–4.5 wt%.

Fixing redox conditions at ΔFMQ-1.5 and the reported range of melt water content, the pre-eruptive average temperature varies between 847 and 857 ± 11 °C, if redox conditions are fixed at ΔFMQ-0.5 and the temperature is in the range 808–815 ± 11 °C. The difference in the average temperature obtained from the olivine of Attalora and Cala delle Giache is in the range of 5–7 °C. A similar exercise can be conducted considering the equation proposed by Di Carlo et al. [26] and the XFe of clinopyroxenes. For pre-GT samples, temperature values obtained were 977 ± 64 and 1021 ± 95 °C, much higher with respect to those obtained previously; on the contrary, the temperature for the post-GT sample was 736 ± 8, fixing redox conditions at ΔFMQ-0.5, and 710 ± 8 °C at ΔFMQ-0.5. This difference, beside the intrinsic variation in clinopyroxene composition, probably reflects the calibration range of the Di Carlo et al. [26] equation, which is calibrated for a XFe in the range 0.80–0.98; hence, we consider those temperature values obtained for pantelleritic samples. The different mineral assemblages and the different results obtained from empirical and thermodynamic models cited above suggest that pre-GT comenditic–pantelleritic magma evolved in a temperature range of 800–850 °C, while post-Green Tuff pantellerite evolved at 700–750°. It is interesting to compare with the nine ignimbrite eruptions, which, although they evolved along a similar liquid line of descent, suggest that the older ones (trachytes–comendites) equilibrated at slightly higher pressures and more oxidizing conditions [3], similar to what is seen here, although in a limited data set, with the three investigated centers.

## 7. Conclusions

We studied three local volcanic centers of different age, two of them (Attalora and Giache) lying along the old La Vecchia caldera (age 140 ka) rim, the third set above the younger Cinque Denti caldera rim (age ≥ 46 ka). We may reassume results as follows:

1. The Patite volcanic center (age ~14 ka) is the sole among the three centers to have erupted clearly pantellerite magma (either as pumice fallout or as lava flows compositionally very similar to the most evolved post-GT pantellerites).
2. The ~69 ka Attalora volcanic center located along the La Vecchia caldera rim and consisted of less evolved comenditic trachytes to comendites, similar to the 85 ka Mordomo ignimbrite.
3. The local Giache volcanic center (age 125–105 ka) is more enigmatic, but due to its position very close to the La Vecchia caldera rim, it likely drained less peralkaline magma, similarly to the Cuddia Attalora center, along the La Vecchia faults. This center represents the resumption of volcanism after the eruption of the Cinque Denti ignimbrite (125 ka), with this latter sharing rather similar $SiO_2$ content, peralkalinity index, and mineralogy [1].
4. Different mineral assemblages and different results of empirical and thermodynamic models [16,20–22] obtained for the products of the three centers suggest that pre-GT comenditic pantelleritic magma evolved in a temperature range of 800–850 °C, while post-Green Tuff pantellerites evolved at lower temperatures of around 700–750 °C, consistent with pre-eruptive temperatures derived by experimental phase equilibria.

## 8. Suggestions for Future Research

Although the volcanic island of Pantelleria has a long and continuous history of alkaline volcanism, there is still much to know about magma generation, storage conditions, and petrogenesis in the systems that predate the Green Tuff. Although recent work [1–3,7] has greatly advanced our understanding of the volcanology, stratigraphy, and geochronology of the pre-Green Tuff ignimbrites, much more petrographic and geochemical work needs to be performed to better understand their petrogenesis, as well as the relationships between trachyte, comendite, and pantellerite both on the island and in the alkalic system in general. However, even with what work has been conducted on these ignimbrites,

much less is known about the dozens of [2] minor volcanic centers such as Attalora and Giache that developed after these eruptions. Indeed, what little data does exist for these centers are presented in this paper. For both Attalora and Giache, as well as these, we hope that this paper may inspire future work that focuses on the petrology, geochemistry, and petrogenesis of intra-ignimbrite volcanic centers on Pantelleria, not just to help more fully-characterize the volcanic history of the island, but to enable us to better understand petrogenetic relationships between various alkalic rocks and to understand the timescales of magmatic evolution.

**Supplementary Materials:** The following are available online at https://www.mdpi.com/article/10.3390/min12040406/s1, Table S1: Bulk Rock Analyses, normalized anhydrous, Table S2: Alkali Feldspar Analyses, Table S3: Clinopyroxene Analyses, Table S4: Olivine Analyses, Table S5: Fe-Ti Oxide Analyses.

**Author Contributions:** Conceptualization, J.C.W., N.J.J., P.R. and S.G.R.; software, J.C.W.; investigation, E.V., J.C.W., N.J.J., S.G.R. and P.R.; data curation, N.J.J., P.F., G.D.G. and R.C.; writing—original draft, J.C.W., P.R. and S.G.R.; writing—review and editing: J.C.W., P.R. and S.G.R. All authors have read and agreed to the published version of the manuscript.

**Funding:** This research received no external funding.

**Data Availability Statement:** Data are included within the article.

**Acknowledgments:** This paper is dedicated to the memory of Nancy Romengo, who enthusiastically studied some of the areas presented here during her MSc thesis. Authors are grateful to two anonymous reviewers and the Academic Editor for their helpful suggestions.

**Conflicts of Interest:** The authors declare no conflict of interest.

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
