# Peer review of "Contrasting Styles of Inter-Caldera Volcanism in a Peralkaline System: Case Studies from Pantelleria (Sicily Channel, Italy)"

_minerals, doi:10.3390/min12040406_

Round 1

Reviewer 1 Report

This manuscript is quite interesting, although it is not clear what the relationship is between this text and what is reported in the following articles:

1 . Rotolo, S. G., Scaillet, S., Speranza, F., White, J. C., Williams, R., & Jordan, N. J. (2021). Volcanological evolution of Pantelleria Island (Strait of Sicily) peralkaline volcano: a review. Comptes Rendus. Géoscience, 353(S2), 1-22.

2. Jordan, N. J., White, J. C., Macdonald, R., & Rotolo, S. G. (2021). Evolution of the magma system of Pantelleria (Italy) from 190 ka to present. Comptes Rendus. Géoscience, 353(S2), 1-17.

3. Stabile, P., Arzilli, F., & Carroll, M. R. (2021). Crystallization of peralkaline rhyolitic magmas: pre-and syn-eruptive conditions of the Pantelleria system. Comptes Rendus. Géoscience, 353(S2), 1-20.

where the volcanological aspect has been quite stressed.

Probably the area studied is different, but how does it collocate in the context of the above three publications?

What is the additional contribution to the previous knowledge regarding the volcanology and petrology of Pantelleria?

The cited bibliography lacks Stabile et al, 2021, why?

Add cationic recalculation for minerals

The authors need to explain these aspects.

Author Response

Thanks for your comments and suggestions

- In the introduction a sentence was added about the importance of the study of these three eruptive centres. The other question about the collocation among the three papers cited, these centers were not studied before (a sentence was added in the abstract to remark this fact), and we wish to offer a basic field volcanological information coupled to whole rock and chemistry, leaving for the future more specific studies.

- Was added the reference (Stabile et al., 2021) as requested

- We added cationic recalculation of mineral analyses in Tables 2 to 5, as requested.

Reviewer 2 Report

This work presents interesting new data that may significantly contribute to a better understanding of peralkalinr magmatism. This contribution is to be added to a number of previous works, partly done by some of the authos of the presented manuscript. In general, previous information is well separated from new data and findings.

Going to criticisms, the first (major) one is that there is no geological map or stratigraphic column., so that the field description is difficult to follow. Including both map and column should be compulsory in any revised version of the manuscript, no matter if original or taken from previous words -just to help the reader to understand descriptions and sampling.

A second, major criticism is that no discussion on trace element geochemistry is provided, in spite the fact that the corresponding data are shown in the table of analyses (I believe that both major and trace element data are new). One would like to compare the trace element contents just as the authors have done with major elements, i.e., plotting them together with those in the previous literature. This could help to appreciate if the authors' interpretations are sound or not. To me, a more complete geochemical interpretation, also interpreting the available trace element evidence, should be also compulsory.

In contrast, my suggestion is to reduce to a minimum (or just drop) the authors' discussion on lava temperature on a geochemical and/or mineral geothermometer basis. As it now stands, this discussion is too speculative, probably because it is too summarized. To me, dealing consistently with this issue would require a much longer work; but in this case, the final rersult would look like two juxtaposed works.

Accordingly, my advice to the authors would be just focus on Petrology, Geochemistry and the time evolution of alkaline magma in Pantelleria. After that, magma physics, including the determination of temperature (why not other physical parameters, too?) in the light of the experimental evidence is an interesting issue that deserves a different work, as well as a much more detailed discussion.

Author Response

Thanks for your comments and suggestions

 We added:

- a new Figure 2 with a general stratigraphic scheme of all the units of the island, as requested

- a new Figure 4 detailed stratigraphic columns of the 3 sites.

We enlarged the discussion sectionRegarding the request of a geological map of the island, we prefer to avoid providing one  because all the key information for the reader are included in the two new figures, and also because the available ones (e.g. Mahood and Hildreth, 1986) are outdated and a new geological map of the island is in progress by our group.

Reviewer 3 Report

This manuscript is well written and presented but it requieres moderate/ major revision before being accepted for publications in Minerals. The main issue is that the discussion is not clear. I find it too vague, it needs much more detail and quantification. I have annotated all my comments on the attached pdf copy

Author Response

We particularly appreciated this review. We made all of the suggested modifications to the text and pdf file.

To Fig 1 was added the legend for normal faults, which was missing.

We enlarged the discussion focusing on all the requested elements (e.g. trace element discussion) and added a new, final section ‘suggestions for future research’, and a sentence at the end of discussion to address at least partially to the question raised at line 319.

Round 2

Reviewer 1 Report

This manuscript is acceptable.

Reviewer 2 Report

In this updated version, the authors have provided mos of the descriptive items I requested. Therefore, I find that the manuscript can be accepted as it now stands.

Reviewer 3 Report

All my comments and suggestions have been taken into account by the authors to prepare the revision of their original manuscript. New interesting information and figures have been added, which help to complement the previous information and to present a most robust study. I find the manuscript acceptable as it is for publication in Minerals